# Solute Carrier Family 29A1 Mediates In Vitro Resistance to Azacitidine in Acute Myeloid Leukemia Cell Lines

**DOI:** 10.3390/ijms24043553

**Published:** 2023-02-10

**Authors:** Monika M. Kutyna, Sophie Loone, Verity A. Saunders, Deborah L. White, Chung H. Kok, Devendra K. Hiwase

**Affiliations:** 1Adelaide Medical School, Faculty of Health and Medical Sciences, University of Adelaide, Adelaide, SA 5000, Australia; 2Precision Cancer Medicine Theme, South Australian Health and Medical Research Institute, Adelaide, SA 5000, Australia; 3Department of Haematology, Royal Adelaide Hospital, Adelaide, SA 5000, Australia; 4Centre for Cancer Biology, University of South Australia and SA Pathology, Adelaide, SA 5000, Australia

**Keywords:** azacitidine, SLC29A1, acquired/secondary resistance, leukemia, AML, MDS, cytotoxicity

## Abstract

Azacitidine (AZA) is commonly used hypomethylating agent for higher risk myelodysplastic syndromes and acute myeloid leukemia (AML). Although some patients achieve remission, eventually most patients fail AZA therapy. Comprehensive analysis of intracellular uptake and retention (IUR) of carbon-labeled AZA (^14^C-AZA), gene expression, transporter pump activity with or without inhibitors, and cytotoxicity in naïve and resistant cell lines provided insight into the mechanism of AZA resistance. AML cell lines were exposed to increasing concentrations of AZA to create resistant clones. ^14^C-AZA IUR was significantly lower in MOLM-13- (1.65 ± 0.08 ng vs. 5.79 ± 0.18 ng; *p* < 0.0001) and SKM-1- (1.10 ± 0.08 vs. 5.08 ± 0.26 ng; *p* < 0.0001) resistant cells compared to respective parental cells. Importantly, ^14^C-AZA IUR progressively reduced with downregulation of *SLC29A1* expression in MOLM-13- and SKM-1-resistant cells. Furthermore, nitrobenzyl mercaptopurine riboside, an SLC29A inhibitor, reduced ^14^C-AZA IUR in MOLM-13 (5.79 ± 0.18 vs. 2.07 ± 0.23, *p* < 0.0001) and SKM-1-naive cells (5.08 ± 2.59 vs. 1.39 ± 0.19, *p* = 0.0002) and reduced efficacy of AZA. As the expression of cellular efflux pumps such as ABCB1 and ABCG2 did not change in AZA-resistant cells, they are unlikely contribute to AZA resistance. Therefore, the current study provides a causal link between in vitro AZA resistance and downregulation of cellular influx transporter *SLC29A1*.

## 1. Introduction

In hematological malignancies, aberrant DNA methylation at CpG islands, areas with high concentrations of cytosine/guanine dinucleotide, leads to the silencing of critical tumor suppressor genes involved in cancer-related pathways [1,2]. Hence, epigenetic modifications, such as DNA methylation, represent an important therapeutic target in hematological malignancies [1,2]. The first generation of DNA hypomethylating agent (HMA) azacitidine (5-azacitidine; AZA), an analogue of the nucleoside cytidine, was developed as conventional cytostatic therapy [3]. At high doses, it was found to be too toxic for patients in the absence of substantial antitumor effect. However, at lower concentration with repeated doses, this HMA was shown to be effective in patients with myelodysplastic syndromes (MDS) [4], a heterogenous group of clonal hematopoietic malignancies characterized by ineffective hematopoiesis, cytopenia, dysplastic features, cytogenetic and molecular abnormalities, and risk of progression to acute myeloid leukemia (AML). These encouraging results led to its approval by the US Food and Drug Administration (FDA) for the treatment of MDS. Though AZA is broadly used for the treatment of MDS [5] and older, medically non-fit AML patients [6], it improved median OS by only 9 months in higher risk MDS (15 months in conventional care vs. 24 months in AZA group) [4] and 5 months in AML (6.9 months in conventional care group vs. 12.1 months in AZA group) [6]. Moreover, despite initial responses to AZA in a subset of MDS patients, the development of resistance to HMA therapy was an inevitable problem [7].

Although the exact mechanism of HMA resistance is not well known, it is broadly classified as primary and secondary resistance. Lack of response after initial 4–6 cycles of AZA is considered primary resistance, while secondary resistance is defined as loss of response after initial response [8]. Cell intrinsic and extrinsic factors such as bone marrow microenvironment play an important role in primary and secondary resistance [8]; however, the contribution of individual factors is not well known.

Importantly, AZA must be transported into cells and phosphorylated before it can be incorporated into DNA and inactivate DNA methyltransferases [9]. Due to the hydrophilic nature of AZA, transport across the plasma membrane by simple diffusion is limited. Solute carrier (SLC) transporters, including SLC28 and SLC29, are well known cellular transporters of multiple nucleoside and nucleoside analogues [10] and therefore were considered potential candidates for AZA transport. However, the role of these transporters in AZA resistance is actively debated as some studies support the role of SLC29A [11,12,13], while others refute it [14,15,16,17,18,19]. Similarly, the role intracellular metabolic pathways such as uridine/cytidine kinase (UCK) [13,14,15,17,20,21,22] and cytidine deaminase (CDA) [15,16,18] remains controversial. This study aims to provide insight into the mechanism of acquired AZA resistance.

## 2. Results

### 2.1. Generation and Characterization of AZA Resistant Cell Lines

In order to understand the mechanism of AZA resistance, MOLM-13 (M) and SKM-1 (S) were exposed in vitro to incrementally increasing concentrations of AZA, beginning at 0.1 µM until overtly resistant to AZA (Appendix A). After establishing cell lines, AZA resistance was evaluated by measuring cell death with increasing concentration of AZA. The concentration of AZA required to induce cell death in 50% of cells in culture was defined as LD_50_. In MOLM-13 AZA-resistant (R) lines, LD_50_ increased by three-fold (4.25 ± 0.25 μM vs. 13.35 ± 1.77 μM; *p* < 0.007) in M-R1 and five-fold (4.25 ± 0.25 μM vs. 20.75 ± 2.67 μM; *p* < 0.003) in M-R5 compared to naïve cells (M-naïve) (Figure 1A). Similarly, LD_50_ was significantly higher in SKM-1-resistant cells (S-R5) compared to naïve cells (S-naïve) (18.72 ± 1.52 μM vs. 5.53 ± 0.34 μM; *p* < 0.001) (Figure 1B). These results confirm that the generated cell lines were resistant to AZA.

Metaphase cytogenetic analyses of MOLM-13- and SKM-1-naïve and AZA-resistant cell lines after being maintained in culture for approximately 5 months were performed. All AZA-resistant MOLM-13 cell lines retained the same multiple numerical and structural aberrations as MOLM-13-naïve (Appendix A). Whereas all SKM-1 AZA-resistant cell lines exhibited a clone with the same complex abnormalities as naïve cells but also presented an additional subclone with del 1q (Appendix A).

Collectively, these results indicated that continuous exposure to AZA induces resistance in vitro. In order to understand the mechanism of AZA resistance, the AZA cellular transport and metabolic activation of the prodrug prior to incorporation into nucleic acids and inhibition of DNMT (thereby inducing DNA methylation) were investigated.

### 2.2. Resistance to AZA Is Due to Reduced Intracellular Uptake and Retention within the Cells

As cellular uptake is the first critical step of AZA resistance, cellular uptake and retention (IUR) using carbon-labeled AZA (^14^C-AZA) in 2 × 10^5^ naïve and AZA-resistant MOLM-13 and SKM-1 cells were evaluated.

Significant reduction in ^14^C-AZA IUR in all MOLM-13 AZA-resistant cell lines (M-R0.4 (3.35 ± 0.24 ng), M-R1 (3.00 ± 0.54 ng), and M-R5 (1.65 ± 0.08 ng) compared to M-naïve (5.79 ± 0.18 ng; *p* < 0.0001)) was observed (Appendix A and Figure 1C). Strikingly, a progressive drop in ^14^C-AZA IUR with increasing resistance to AZA was observed. For example, ^14^C-AZA IUR was significantly lower in M-R5 compared to M-R1 (1.65 ± 0.08 vs. 3.00 ± 0.54 ng; *p* < 0.0001) (Figure 1C). These findings were validated in other AML cell line, SKM-1. The ^14^C-AZA IUR was significantly lower in the SKM-1-resistant cell line S-R1 (3.37 ± 0.32 vs. 5.08 ± 0.26 ng; *p* < 0.001) and S-R5 (1.10 ± 0.08 vs. 5.08 ± 0.26 ng; *p* < 0.0001) (Figure 1D) but not in S-R0.4 µM AZA (Appendix A) compared to S-naïve. Significantly, a higher LD_50_ in the AZA-resistant MOLM-13 and SKM-1 cell lines (Figure 1A,B) correlated with a reduced amount of AZA in the cells (Figure 1C,D). Collectively, these findings indicated that in vitro AZA resistance was driven by low intracellular concentration of AZA.

Next, the mechanism of lower IUR in resistant cells was investigated. Intracellular concentration is a dynamic process and is a net balance of cellular influx and efflux. Cellular influx is a combination of passive uptake or diffusion across the concentration gradient, and energy dependent active uptake. Passive cellular uptake is not temperature dependent, whereas active cellular transport is temperature dependent, and hence the ^14^C-AZA IUR in the naïve and resistant cells at 37 °C and 4 °C was assessed. Strikingly, ^14^C-AZA IUR was significantly lower at 4 °C compared to 37 °C in MOLM-13 naïve cells (*p* < 0.0001) (Figure 2A). However, there was no such difference in highly resistant cells (Appendix A). Together these findings suggest that ^14^C-AZA uptake in naïve cells was partly mediated by influx pump, which was influenced by temperature of the culture conditions. The thermal effect of ^14^C-AZA IUR in highly resistant cells could be due to reduced expression and/or activity of the temperature-dependent active pump.

### 2.3. Differential Expression of Nucleoside Transporters in AZA-Resistant Cells

Gene expression profile of 27 drug transporters in MOLM-13, M-R0.4. M-R1, M-R5 was performed. Principal component analysis (PCA) based on gene expression indicated that all AZA-resistant cell lines have a distinct gene expression profile compared to naïve control (Figure 2B). Interestingly, among the resistant cells, M-R0.4 and M-R1 samples were clustered closer to each other compared to M-R5. This indicated that the gene expression pattern in M-R0.4 and M-R1 cell lines were similar to each other compared to M-R5, which had distinct gene expression pattern (Figure 2B). As shown in Figure 2C, the majority of the genes differentially expressed between naïve and resistant cells were members of the nucleoside or ATP-binding cassette (ABC) transporter families (FDR *p* < 0.05). *SLC29A1* was the most highly expressed SLC gene in MOLM-13 naïve cells (Appendix A), with similar expression observed in SKM-1-naïve cells (Appendix A). Moreover, expression of *SLC29A1*, *SLC29A2*, *SLC29A4*, *SLC19A1*, and *SLC25A13* were significantly downregulated in resistant cells compared to parental cells (Figure 2D,E, Appendix A). Downregulation of *SLC29A1* was also observed in SKM-1-resistant cells (Figure 2F and Appendix A), while expression of *SLC29A2* was unchanged (Appendix A).

### 2.4. In Vitro AZA Cellular Uptake Is SLC29A1 Dependent

The ^14^C-AZA IUR progressively reduced as *SLC29A1* and *SLC29A2* expression reduced in MOLM-13-resistant cells (Figure 3A). These findings suggested that *SLC29A1* and/or *SLC29A2* play significant role in cellular influx of ^14^C-AZA.

Nitrobenzyl mercaptopurine riboside (NBMPR), an SLC29A inhibitor (Appendix A), reduced AZA IUR by almost three-fold in MOLM-13-naïve cells (5.79 ± 0.18 vs. 2.07 ± 0.23, *p* < 0.0001) (Figure 3B). Interestingly, NBMPR reduced ^14^C-AZA IUR in cell lines resistant to lower concentration of AZA but not highly resistant cells. For example, NBMPR reduced ^14^C-AZA IUR in M-R0.4 (3.35 ± 0.24 vs. 1.58 ± 0.28 ng, *p* = 0.002) and M-R1 cells (3.21 ± 0.61 vs. 1.18 ± 0.11 ng, *p* = 0.044) but not in M-R5 (1.65 ± 0.08 vs. 1.54 ± 0.18 ng, *p* = 0.665) (Figure 3B). The effect of NBMPR on ^14^C-AZA IUR was replicated in SKM-1-naïve and -resistant cells (Figure 3C). Together these results provided evidence that NBMPR blocks cellular transporter(s) which are involved in ^14^C-AZA cellular influx.

Strikingly, co-culture with NMBR reduced efficacy of AZA leading to higher LD_50_ of AZA MOLM-13 (6.03 ± 0.45 vs. 4.25 ± 0.25 µM; *p* = 0.025) and SKM-1-naïve cells (7.74 ± 0.34 vs. 5.53 ± 0.34 µM; *p* = 0.010) (Figure 3D).

Collectively, these results demonstrated that AZA resistance is mediated by reduced intracellular concentration which in turn is predominantly driven by reduced expression of *SLC29A1* and *SLC29A2*.

### 2.5. Resistance to AZA Is Not Mediated by ABCB1 and ABCG2

ABCB1 and ABCG2 are common drug efflux proteins reported to mediate resistance to multiple chemotherapeutics agents. However, ABCB1 and ABCG2 surface protein expression did not change in AZA-resistant cells (Figure 4A–D). Furthermore, ^14^C-AZA IUR in ABCG2-overexpressing leukemia cells (K562-ABCG2B) did not alter with ABCG2 inhibitor KO143 (12.14 ± 1.01 ng vs. 12.24 ± 1.09 ng, *p* = 0.922) (Figure 4E).

Similarly, cyclosporine, an ABCB1 inhibitor did not influence the ^14^C-AZA IUR in an ABCB1 over-expressing cell line (K562-DOX) (10.84 ± 1.68 ng vs. 11.66 ± 1.42 ng, *p* = 0.723), MOLM-13-naïve (5.81 ± 0.29 ng vs. 4.80 ± 0.94 ng, *p* = 0.209) or MOLM-13 AZA-resistant cells (1.65 ± 0.09 ng vs. 1.65 ± 0.32 ng, *p* = 0.978) (Figure 4F,G). Collectively, these results suggested that ABCB1 and ABCG2 do not mediate AZA efflux and hence are unlikely to mediate AZA resistance in vitro.

The role of other efflux transporters in AZA cellular efflux is not well known. Based on transporter genes expression profiling, *ABCB4* expression progressively increased in resistant cells compared to MOLM-13-naïve cells (Appendix A). Similarly, *ABCB4* expression also increased in SKM-1-resistant cells (Appendix A). An ABCB4 inhibitor, verapamil (Appendix A), did not alter ^14^C-AZA IUR in MOLM-13- and SKM-1-resistant cell lines (*p* = 0.558 and *p* = 0.330, respectively) (Appendix A). Taken together, there was no strong evidence that ABC transporters (ABCB1, ABCG2, ABCB4) play a significant role in contributing to cellular transport of ^14^C-AZA.

In order to delineate the effect of other transporters in ^14^C-AZA influx/efflux, a panel of drugs used in routine clinical practice that can inhibit activity of various transporters was tested. A list of the drugs and their targets are summarized in Appendix A. *SLC22A1* and *SLC22A2* in MOLM-13 and SKM-1 showed minimal change in expression between naïve and AZA-resistant cells. This was further supported by inhibitors studies. For instance, chloroquine, amantadine, and SLC22A1 and SLC22A2 inhibitors reduced ^14^C-AZA IUR in MOLM-13-naïve and M-R0.4 but not in M-R1 and M-R5 cells (Appendix A). Although a similar trend was observed in SKM-1 cell lines, the differences were not statistically significant (Appendix A). Unexpectedly, amantadine but not chloroquine increased sensitivity of AZA, assessed by LD_50_, in MOLM-13 cells and SKM-1 cells (Appendix A). Other transporter inhibitors, such as procainamide, corticosterone, cimetidine, and pyrimethamine did not influence the ^14^C-AZA IUR in either naïve or R5 AZA-resistant MOLM-13 and SKM-1 cell lines (Appendix A). Together these findings suggested that SLC22A1, SLC22A2, and ABC transporters are less likely involved in mediating AZA resistance.

### 2.6. Deregulation of AZA Metabolism Genes in AZA Resistant Cell Lines

Upon AZA transportation into cells, activation of the prodrug AZA is necessary. The change in expression of genes mediating AZA phosphorylation, *UCK1*, and *UCK2*, in resistant cells was assessed. There were no changes in mRNA expression of either *UCK1* or *UCK2* in MOLM-13 AZA-resistant cell lines compared to naïve (Figure 5A,B).

However, expression of *CDA*, a mediator of AZA deamination, was elevated in intermediate resistant cells (R0.4 and R1) but downregulated in highly resistant cells MOLM-13 and SKM-1 cells compared to naïve cells (Figure 5C,D).

## 3. Discussion

Azacitidine is the most widely used HMA for management of MDS and older AML patients either as single agent or in combination with venetoclax. Though treatment failure is inevitable, the mechanisms of AZA resistance are not well known. The current study provides mechanistic data linking in vitro AZA resistance and downregulation of the cellular influx pathways. Key findings of the present study include: (i) AZA cellular uptake in leukemia cell lines was predominantly mediated by SLC29A1; (ii) AZA resistance induced by in vitro exposure to escalating doses of AZA was mediated by reduced intracellular concentration; (iii) reduced intracellular concentration of AZA in resistant cells was mediated by downregulation of *SLC29A1*; (iv) cellular efflux pumps such as ABCB1 and ABCG2 did not contribute to AZA resistance; (v) *CDA* expression in AZA-resistant cells was dynamic, with a progressive increase in *CDA* expression in intermediate resistant cells followed by sharp reduction in cells resistant to higher doses of AZA.

The current study demonstrated that SLC29A1 mediates AZA uptake in leukemia cell lines and, importantly, provided mechanistic data linking AZA resistance with lower intracellular concentration of AZA, which is predominantly mediated by reduced expression of *SLC29A1* in resistant cells. Compared to other SLCs, *SLC29A1* expression was much higher in the naïve cell lines examined in this study. Similarly, high expression of *SLC29A1* was observed in primary leukemia blast and other human leukemia cell lines [11]. Furthermore, current study demonstrated that *SLC29A1* expression directly correlated with AZA intracellular concentration, which is consistent with high AZA cellular uptake in primary marrow blasts, and leukemia cell lines transfected with *SLC29A1*, with lower uptake in normal fibroblast cells expressing very low levels of *SLC29A1* [11]. CRISPR/Cas9 knockout screen and oligonucleotide arrays also suggested direct correlation between *SLC29A1* expression and potency of AZA in human cancer cell lines [12,13]. Furthermore, in the present study, the SLC29A1 inhibitor significantly reduced AZA intracellular concentration in naïve cells but not in resistant cells. Reduced AZA intracellular concentration in primary bone marrow blasts and other leukemia cell lines by SLC29A1 inhibition [11,12] led to reduced cytotoxicity and DNA methylation [11]. Collectively, these findings suggested that endogenous high expression of *SLC29A1* plays a critical role in AZA uptake in primary AML blasts and leukemia cell lines [11]. However, these findings are not without conjecture. Other studies could not show correlation between *SLC29A1* expression and the IC50 of AZA in AML and human cancer cell lines [15,16]. The link between acquired AZA resistance and lower intracellular concentration was also questioned. The intracellular level of AZA in resistant and parental cells were similar in some studies, indicating that reduced cellular uptake is less likely to be the molecular mechanism of underlying acquired resistance [17] and reduced expression in *SLC29A1* could not be demonstrated in resistant cells [14,18,19]. Similarly, conflicting results of *SLC29A1* expression in MDS patients treated with AZA or decitabine were reported. Significantly higher *SLC29A1* mRNA expression was observed in decitabine responders compared to non-responders [23,24]. While in other studies such a correlation was not observed [25,26]. Moreover, there was no change in expression at relapse [24,25]. It is noteworthy that the majority of these studies included patients treated with decitabine [23,24,25] with only one study of AZA-treated patients [26].

In vitro studies employing model of forced overexpression of SLC in Saccharomyces cerevisiae and Xenopus oocyte suggested that all seven SLCs (SLC28A1-3 and SLC29A1-4) transported AZA [27] with strong interaction with SLC28A3 and SLC28A1 [27]. This was supported by increased AZA sensitivity of canine kidney and leukemia cell lines transfected with *SLC28A1* [27,28,29] compared to cells lines transfected with *SLC29A1*, *SLC29A2*, and *SLC28A2* [27]. Collectively these studies demonstrated that SLC28A1 and SLC28A3 exhibited stronger interaction with AZA. However, in the current study, endogenous expression of *SLC28A1* and *SLC28A2* was not detectable, while *SLC28A3* expression was very low. In line with the present findings, *SLC28A1*, *SLC28A2*, and *SLC28A3* were poorly expressed in primary bone marrow blasts and leukemia cell lines [11]. Moreover, AZA cellular uptake is Na^+^-independent, while SLC28As-mediated cellular uptake is highly Na^+^-dependent [11]. Collectively, these results suggested that SLC28As are unlikely to be predominant carrier of AZA in leukemia cells.

Cellular transporters ABCB1 (MDR1) and ABCG2 function as efflux pumps with broad specificities. They are highly expressed in many human cancers, including leukemia, and confer resistance, poor response to therapy, and survival (reviewed in [30]). Multiple findings of the current study demonstrated that AZA is not effluxed by ABCB1 or ABCG2. In vitro exposure to escalating concentration of AZA did not lead to overexpression of these efflux pumps. Secondly, ABCG2 and ABCB1 inhibitors did not alter AZA intracellular concentration in ABCG2- and ABCB1-overexpressing cells, respectively. In agreement with the present findings, other groups also reported a lack of increased *ABCB1* [14,19,20,31], *ABCC1*, and *ABCG2* [31] expression in AZA-resistant cells compared to their parental cells and in bone marrow blasts from AZA-treated MDS patients at the time of relapse [25]. In contrast to these findings, continuous exposure to AZA induced *ABCB1* expression that conferred resistance to other ABCB1 substrates [32] but not to AZA. Collectively, the current and other groups results demonstrated that AZA is not a substrate for ABCB1 or ABCG2, and AZA resistance is not mediated by ABCB1 and ABCG2. However, the role of other ABC transporters in AZA resistance is not well known.

Once AZA is transported into cells, activation of the prodrug is necessary. The first rate-limiting step is the phosphorylation to the AZA-monophosphate, which is mediated by UCK1 and UCK2. There were no significant changes in *UCK1* and *UCK2* expression in AZA-resistant cells compared to their parental cells. Similarly, UCK1 and UCK2 protein expression did not correlate with the cytotoxic effect of the AZA in AML cell lines [15], and their expression did not change in resistant cells compared to their parental cells [14,15]. In MDS patients treated with AZA, *UCK2* expression was not different in responders versus non responders [21]. However, in a panel of 60 cancer cell lines, AZA sensitivity correlated with *UCK2* but not *UCK1* expression [22]. Genome-wide CRISPR/Cas9 knockout screen identified *UCK2*, but not *UCK1*, as a rate-limiting enzyme for AZA activation [13]. *UCK2* mRNA and protein expression decreased in resistant cells compared to their parental cells [17,20], and acquired mutation reduced UCK2 activity [14]. Furthermore, downregulation of *UCK2* expression was observed at relapse of AZA-treated MDS patients [21]. Collectively, correlation between *UCK 1* and *2* expression and its causal relation with primary and secondary AZA resistance is actively contested.

After phosphorylation, 80–90% of AZA is incorporated into RNA, while 10–20% is incorporated into DNA [9,33], and enhanced AZA incorporation into RNA is associated with worse treatment response [34]. Elimination of AZA on the other hand occurs by deamination with CDA [35]. In the current study *CDA* expression was highly dynamic, progressively increased in intermediate resistant cells (resistant to 0.4 and 1 µM of AZA) but reduced in cells resistant to 5 µM AZA. Similarly, protein expression increased in AZA-resistant AML [15] and colorectal cancer cell lines [18] compared to their parental cells. However, other studies did not find correlation between mRNA [16] and protein [15] expression of CDA and IC50 of AZA [15,16]. In another study, though, *CDA* and *DCK* expression were not different, and the *CDA/DCK* ratio was three-fold higher in patients responding to decitabine [25]; however, these findings could not be validated [23,24]. Similarly, changes in *CDA* mRNA and the *CDA*/*DCK* ratio were not observed at the time of relapse [24,25]. Plasma CDA levels were reported to be high in men compared to women and were correlated with poor survival [36] in males; however, these findings could not be validated in other studies [37,38]. Further research is required to define causal link between *CDA* expression and primary and secondary AZA resistance.

In summary, the current study demonstrated that in vitro secondary AZA resistance was mediated by reduced intracellular concentration of AZA which in turn was mediated by downregulation of *SLC29A1* expression. If validated in primary patient samples, *SLC29A1* expression could be used as a biomarker to predict response to AZA. Moreover, therapeutic strategies regulating *SLC29A1* expression and activity can be exploited to improve efficacy of AZA.

## 4. Materials and Methods

### 4.1. Cell Culture

SKM-1 cell line was purchased (DSMZ, Braunschweig, Germany), while MOLM-13, K562-ABCG2 [39], K562, and K562-DOX were regularly maintained in our laboratory. All cell lines were maintained at 37 °C and 5% CO_2_ and cultured in RPMI 1640 supplemented with 10% foetal bovine serum (FBS), 1% penicillin (50 units/mL)/streptomycin (50 µg/mL) and 1% α-glutamine (200 mM) (Sigma-Aldrich, St. Louis, MO, USA) and seeded at 5 × 10^5^ cells/mL every 3–4 days.

### 4.2. Generation of AZA Resistant Cell Lines

Cell lines, MOLM-13 and SKM-1 were exposed to gradually escalating AZA concentrations (kindly supplied by BMS/Celgene, Melbourne, Australia) starting at 0.1 µM. AZA concentration was increased by 0.1–0.5 μM AZA approximately every 10–14 days until reaching 10 μM. Experiments were performed predominately on the cell lines resistant to 0.4 µM (R0.4), 1 µM (R1) and 5 µM AZA (R5). AZA was dissolved in dimethyl sulfoxide (DMSO; Sigma-Aldrich, St. Louis, MO, USA). Control (naïve) cell lines cultured in 0.1% DMSO were maintained in parallel.

### 4.3. Cell Viability Assays

SKM-1- and MOLM-13-naïve and AZA-resistant cells were cultured in a 24-well plate (Thermo Fisher Scientific, Waltham, MA, USA) with increasing concentration of AZA (1 and 5 μM) for 72 h and LD_50_ was assessed by flow cytometry analysis (BD FACS Canto or Fortessa) after staining with 7-aminoactinomycin (7-AAD; Invitrogen Life Technologies, Carlsbad, CA, USA) and phycoerythrin (PE)-conjugated Annexin V (BD Biosciences, Franklin Lakes, NJ, USA).

### 4.4. Intracellular Uptake and Retention Assays (IUR)

IUR [40] were performed in triplicate with 2 × 10^5^ cells per tube. Briefly, 2 × 10^5^ cells were incubated for 2 h at 37 °C in the presence and absence of a 50% ^14^C-labeled 2 μM AZA. For IUR with inhibitors, 100 μM verapamil (Royal Adelaide Hospital Pharmacy, Adelaide, SA, Australia), 200 μM procainamide (Sigma-Aldrich, St. Louis, MO, USA), 10 μM corticosterone (Sigma-Aldrich, St. Louis, MO, USA), 20 μM NBMPR (Sigma-Aldrich, St. Louis, MO, USA), 20 μM cyclosporin A (Sigma-Aldrich, St. Louis, MO, USA), 10 μM chloroquine (Sigma-Aldrich, St. Louis, MO, USA), 150 μM amantadine (Sigma-Aldrich, St. Louis, MO, USA), 20 and 200 μM cimetidine (Sigma-Aldrich, St. Louis, MO, USA), and 0.1 and 10 μM pyrimethamine (Sigma-Aldrich, St. Louis, MO, USA) were added. After incubation the cellular and aqueous phases were separated, and incorporation determined using a Perkin Elmer Liquid Scintillation Analyser following the addition of Microscint 20 scintillation fluid (Perkin Elmer, Waltham, MA, USA) before counts per minute of β radiation in the supernatant and cell pellet fractions was used to convert to ng of AZA in 2 × 10^5^ cells. All assays were performed in triplicate and repeated if the assay demonstrated non-concordance.

### 4.5. Immunophenotyping for ABCB1 and ABCG2

Surface expression of ABCB1 and ABCG2 efflux transporter proteins was assessed by flow cytometry analysis of MOLM-13- and SKM-1-naïve and AZA-resistant cell lines. ABCG2-overexpressing K562-ABCG2B and ABCB1-overexpressing K562-DOX and their parental cells were used as controls for individual experiments. Cells were stained with phycoerythrin-conjugated (PE) anti-ABCB1 antibody (Beckman Coulter, IM2370U) and anti-ABCG2 (R&D Systems, FAB9950). The data were acquired on the BD FACS CANTO II flow cytometry machine and analysed using FlowJo analysis software version 9.

### 4.6. Total RNA Isolation and Quantitative PCR

Total RNA was isolated using TRIzol^®^ (Invitrogen Life Technologies, Carlsbad, CA, USA) followed by complementary DNA (cDNA) synthesis using random hexamers (GeneWorks, Hindmarsh, SA, Australia) and Superscript^®^ II Reverse Transcriptase (Invitrogen Life Technologies).

A 27-gene Taqman^®^ transporter gene assay plate (Thermo Fisher Scientific, Waltham, MA, USA) was designed and carried out according to the manufacturer’s instructions. Quantitative PCR was performed on the QuantStudio 7 (Applied Biosciences, Waltham, MA, USA). Results were analyzed with the QuantStudio 7 instrument as previously described [41]. The raw data were normalized against endogenous control gene TATA-Box Binding Protein (TBP) (Life Technologies, Hs99999910_m1) using the delta-delta Ct (ΔΔCt) method as implemented in the HTqPCR Bioconductor package [42]. Triplicate gene expression values were averaged for each gene for each sample before subsequent analyses. Taqman^®^ FAM-NFQ-MGT labeled primers for *SLC22A1*, *SLC22A2*. *SLC22A3*, *ABCB11*, *SLC2A3*, *ABCC1*, *SLC19A1*, *SLC29A3*, *SLC29A4*, *ABCC4*, *ABCF1*, *VDAC1*, *SLC25A13*, *ABCA2*, *ABCB4*, *ATP7A*, *ATP7B*, *TAP1*, *SLC7A8*, *ABCD1*, *SLCO3A1*, *SLC28A3*, *SLC29A1*, *SLC29A2*, *UCK1*, *UCK2*, and *CDA* (Life Technologies, Hs00427552_ml, Hs01010723_ml, Hs01009568_ml, Hs00184824_m1, Hs00359840_m1, Hs00219905_m1, Hs00953342_m1, Hs00217911_m1, Hs00928283_m1, Hs00988734_m1, Hs00153703_m1, Hs01631624_ml, Hs00185185_m1, Hs00242232_m1, Hs00240956_m1, Hs00163707_m1, He00163739_m1, Hs00388675_m1, Hs00794796_m1, Hs00163610_m1, Hs00203184_m1, Hs00910439_m1, Hs01085706_m1, Hs00155426_m1, Hs01075618_m1, Hs00367072_m1, Hs0015601_m1, respectively) were used in this study.

### 4.7. Cytogenetic Analysis

Conventional assessment using karyotype analysis was performed in Cytogenetics Laboratory at Genetics and Molecular Pathology, SA Pathology, Adelaide, on unselected BM aspirates according to standard methods. Briefly, metaphase cells were spread onto glass slides for G-banding; 35 to 70 metaphase images were collected automatically by the MetaFer scanning system (MetaSystems, New Castle, DE, USA) and a minimum of 20 cells were examined.

### 4.8. Statistical Analysis

Pairwise comparisons were performed using empirical Bayes-moderated t statistics that implemented in limma R package [43] as previously described [41]. The false-discovery rate (FDR) was controlled using the Benjamini–Hochberg algorithm [44]. Principal component analysis (PCA) of the gene expression (ΔCt values) dataset was calculated in R using the prcomp function. Briefly, the data were standardized by subtracting the mean and dividing by the standard deviation for each variable. Then, the covariance matrix was computed to identify correlation. The eigenvectors and eigenvalues of the covariance matrix were computed to identify the principal components (PCs). The data were then projected onto the first two components PC1 and PC2. Samples that have similar gene expression profiles are clustered together. Heatmaps were generated using the pheatmap R package. Volcano plot was performed using ggplot package visualized significant genes and their fold change (−log10(FDR *p*-value) against log2 fold-change). Statistical analysis was performed using Student’s t-test to determine difference between experimental groups. Normality test was performed using GraphPad prism 8 software. Only variable with *p* < 0.05 was considered to be statistically significant. All the analysis and graphs were generated using the GraphPad Prism 8 statistical software (GraphPad Prism Inc., La Jolla, CA, USA) or R statistical software version 4.1.1.

## Figures and Tables

**Figure 1 ijms-24-03553-f001:**
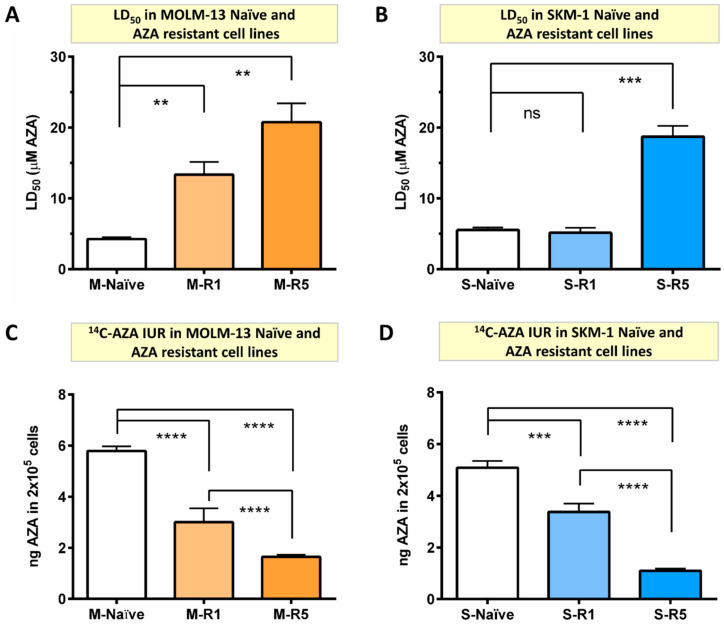
The ^14^C-AZA intracellular uptake and retention (IUR) is significantly lower in AZA-resistant cell lines compared to parental cells. The concentration of AZA required to kill 50% of cells (LD_50_) was determined by Annexin V/7-AAD staining. (**A**) AZA LD_50_ was significantly higher in MOLM-13-resistant cells (M-R1 and M-R5) compared to naïve cells; (**B**) similarly, AZA LD_50_ was significantly higher in SKM-1-resistant cells (S-R5) compared to naïve cells; Importantly, resistance is probably driven by the reduced intracellular concentration of AZA, as shown by reduced ^14^C-AZA IUR in (**C**) MOLM-13- and (**D**) SKM-1-resistant cells compared to their parental cells. Data represents the mean and all error bars are indicative of SEM of at least 3 independent experiments. AZA, azacitidine; M, MOLM-13; S, SKM-1; ns, not significant. Unpaired Student’s *t*-test (Welch’s correction was applied for data groups with unequal SD) was used to detect statistically significant differences between cohorts. Asterisks display *p*-values ** *p* < 0.01, *** *p* < 0.001, **** *p* < 0.0001.

**Figure 2 ijms-24-03553-f002:**
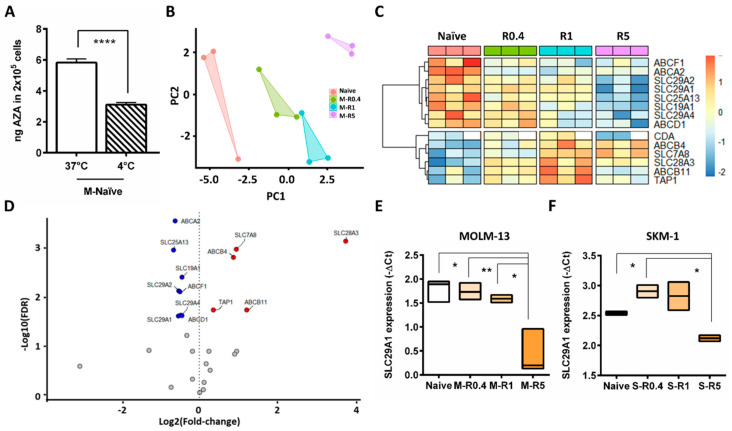
Differential expression of transporter genes in MOLM-13 AZA-resistant compared to parental cells. (**A**) In MOLM-13-naïve cells ^14^C-AZA IUR was significantly lower at 4 °C compared to 37 °C suggesting temperature dependent active cellular transport. (**B**) Principal component analysis (PCA) showed that intermediate resistant cells (M-R0.4 and M-R1), highly resistant cells (M-R5), and naïve cells exhibit distinct gene expression profile. (**C**) Heatmap of the cellular transporter and metabolism genes differentially expressed in naïve and resistant MOLM-13 cells. White color boxes in *CDA* indicate lack of data. (**D**) Volcano plot demonstrating differential gene expression between resistant and naïve MOLM-13 cells. *X*-axis show the effect of log2 fold change while log10-adjusted *p* value (FDR) is shown on the *y*-axis. Red circles indicate increased gene expression in AZA-resistant cells compared to naïve (FDR adjusted *p* < 0.05 and log2 fold change >0), while blue circles indicate genes downregulated in AZA-resistant compared to naïve cells (FDR adjusted *p* < 0.05 and log2 fold change <0). Box plot representing *SLC29A1* expression in (**E**) MOLM-13 and (**F**) SKM-1 cells. Student’s *t*-test was used to detect statistically significant differences between cohorts. Asterisks display *p*-values * *p* < 0.05, ** *p* < 0.01, **** *p* < 0.001.

**Figure 3 ijms-24-03553-f003:**
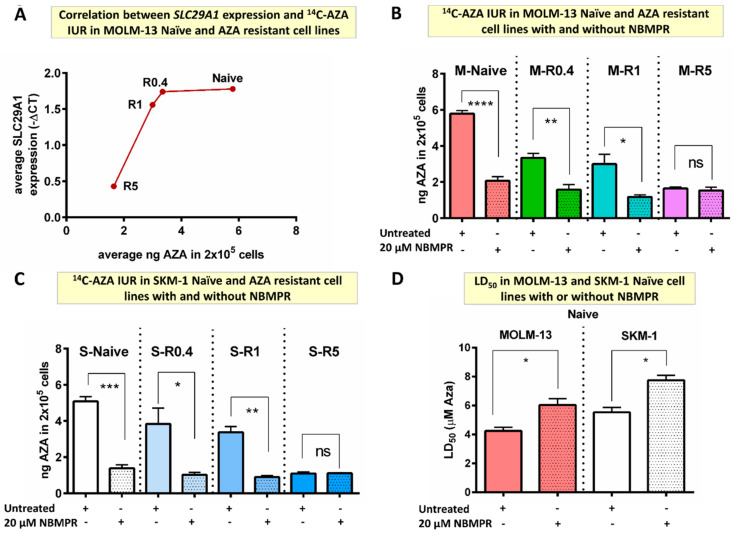
SLC29A inhibitor decreased ^14^C-AZA IUR in MOLM-13- and SKM-1-naïve and AZA-resistant cell lines. (**A**) *SLC29A1* expression was downregulated in resistant cells compared to naïve MOLM-13 cells. Importantly, ^14^C-AZA IUR was inversely corelated to *SLC29A1* expression; NBMPR, an SLC29A inhibitor, reduced ^14^C-AZA IUR in (**B**) MOLM-13- and (**C**) SKM-1-naïve and intermediate resistant (R0.4 and R1) but not in highly resistant (R5) cells; (**D**) NBMPR reduces sensitivity to AZA in MOLM-13- and SKM-1-naïve cells. Data represent the mean and all error bars indicate SEM of at least 3 independent experiments. AZA, azacitidine; M, MOLM-13; S, SKM-1; ns, not significant. Unpaired Student’s *t*-test (Welch’s correction was applied for data groups with unequal SD) was used to detect statistically significant differences between cohorts. Asterisks display *p*-values * *p* < 0.05, ** *p* < 0.01, *** *p* < 0.001, **** *p* < 0.0001.

**Figure 4 ijms-24-03553-f004:**
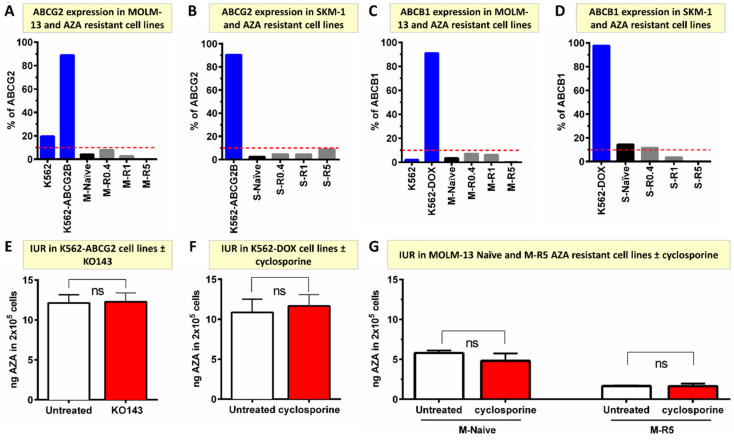
ABCG2 and ABCB1 do not mediate AZA cellular efflux and unlikely contribute to resistance in vitro. (**A**,**B**) ABCG2 protein is not expressed in MOLM-13- and SKM-1-naïve and AZA-resistant cell lines. ABCG2 over-expressing K562-ABCG2 and ABCG2 negative K562 were used as controls for individual experiments; (**C**,**D**) ABCB1 is not expressed in MOLM-13-and SKM-1-naïve and AZA-resistant cell lines. ABCB1 over-expressing K562-DOX and ABCB1 negative K562 were used as controls for individual experiments; (**E**) ABCG2 inhibitor (KO143) did not alter ^14^C-AZA IUR in K562-ABCG2-overexpressing cells; (**F**) ABCB1 inhibitor (cyclosporine) did not influence ^14^C-AZA IUR in K562-DOX cells overexpressing ABCB1; (**G**) cyclosporine did not change ^14^C-AZA IUR in MOLM-13-naïve and -resistant cells. Data represent the mean of at least 3 independent experiments. AZA, azacitidine; M, MOLM-13; S, SKM-1; ns, not significant.

**Figure 5 ijms-24-03553-f005:**
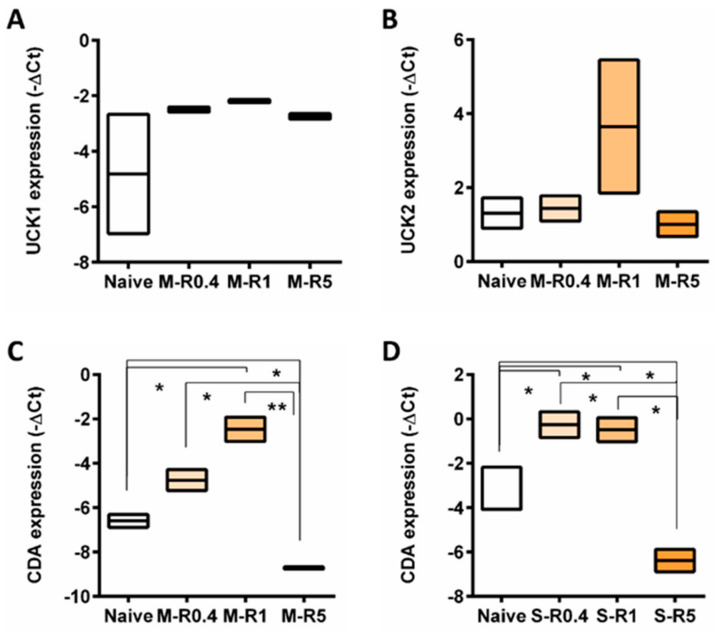
The expression of genes involved in AZA metabolism across naïve and AZA-resistant cell lines. Box plot representing gene expression of (**A**) *UCK1* and (**B**) *UCK2* in MOLM-13, *CDA* in (**C**) MOLM-13 and (**D**) SKM-1 cell lines. Student’s *t*-test was used to detect statistically significant differences between cohorts. Asterisks display *p*-values * *p* < 0.05, ** *p* < 0.01.

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
