# Peer review of "Solute Carrier Family 29A1 Mediates In Vitro Resistance to Azacitidine in Acute Myeloid Leukemia Cell Lines"

_ijms, 2023, doi:10.3390/ijms24043553_

Round 1

Reviewer 1 Report

The authors explore the mechanisms of AZA resistance in in vitro models of AML, with more emphasis on SLC function. The work is very clear, well written, and easy to follow. In my opinion, minor considerations need to be addressed:

- the title should be adjusted to AML cell lines since the authors only used AML models in the manuscript.

- Figure S7 should be integrated into the main manuscript to support section 2.6 of the results. 

Author Response

Point 1: The title should be adjusted to AML cell lines since the authors only used AML models in the manuscript.

Response 1: Thank you for your valid point. We have now modified the title to” Solute Carrier Family 29A1 Mediates in vitro Resistance to Azacitidine in Acute Myeloid Leukemia Cell Lines”.

Point 2: Figure S7 should be integrated into the main manuscript to support section 2.6 of the results. 

Response 2: Thank you for the suggestion. We have now integrated Figure S7 in revised manuscript as a Figure 5.

Reviewer 2 Report

This manuscript deals with an important topic. However, some revisions are required to fit for publication as follows:

1.      The abstract lacks quantitative information about the research methods, and the presentation is confusing. There is a paucity of detail about the experimental protocols and which metrics were calculated. The authors could shorten the background and give more details on the methods used. Also, azacitidine has been given the abbreviation AZA throughout the manuscript. Hence, the same abbreviation should be used in the abstract.

2.      Keywords: more representative keywords should be added, like leukemia, cytotoxicity,….etc

3.      Introduction needs to be more detailed and give informative background on the subject, particularly the earlier studies that investigated azacitidine resistance.

4.      Results: all introductory paragraphs should be transferred to the discussion section (E.g. lines 63-66, 86-88..etc).

5.      Line 385: clarify the tested concentrations.

6.      There is a problem with using abbreviations throughout the manuscript. The full term should be mentioned first with the abbreviation between paresis then the abbreviations should be exclusively used throughout the manuscript. E.g., Line 20: AML should be presented as acute myeloid leukemia (AML), then the abbreviation should be used further. Such errors have been repeated for many abbreviations throughout the manuscript.

7.      The writing style should be formal from the third-person perspective. Do not use we or our (E.g. line 25, our study should be the current study).

8.      It is not preferable to begin sentences with abbreviations like AZA in line 67.

Author Response

Point 1:   The abstract lacks quantitative information about the research methods, and the presentation is confusing. There is a paucity of detail about the experimental protocols and which metrics were calculated. The authors could shorten the background and give more details on the methods used. Also, azacitidine has been given the abbreviation AZA throughout the manuscript. Hence, the same abbreviation should be used in the abstract.

Response 1: We acknowledge your concern and abstract is revised according to your suggestions. It reads as:

Azacitidine (AZA) is commonly used hypomethylating agent for higher risk myelodysplastic syndromes and acute myeloid leukemia (AML). Although some patients achieve remission, eventually most patients fail AZA therapy. Comprehensive analysis of intracellular uptake and retention (IUR) of carbon labelled AZA (14C-AZA), gene expression, transporter pump activity with or without inhibitors, and cytotoxicity in Naïve and resistant cell lines provided insight into the mechanism of AZA resistance. AML cell lines were exposed to increasing concentrations of AZA to create resistant clones. 14C-AZA IUR was significantly lower in MOLM-13 (1.65 ± 0.08 ng vs. 5.79 ± 0.18 ng; P < 0.0001) and SKM-1 (1.10 ± 0.08 vs. 5.08 ± 0.26 ng; P < 0.0001) resistant cells compared to respective parental cells. Importantly, 14C-AZA IUR progressively reduced with downregulation of SLC29A1 expression in MOLM-13 and SKM-1 resistant cells. Furthermore, nitrobenzyl mercaptopurine riboside, a SLC29A inhibitor, reduced 14C-AZA IUR in MOLM-13 (5.79 ± 0.18 vs. 2.07 ± 0.23, P < 0.0001) and SKM-1 Naïve cells (5.08 ± 2.59 vs. 1.39 ± 0.19, P = 0.0002), and reduced efficacy of AZA. As the expression of cellular efflux pumps such as ABCB1 and ABCG2 did not change in AZA resistant cells, they are unlikely contribute to AZA resistance. Therefore, the current study provides causal link between in vitro AZA resistance and downregulation of cellular influx transporter SLC29A1.”

Point 2: Keywords: more representative keywords should be added, like leukemia, cytotoxicity,….etc

Response 2: Thank you for your suggestion. We have now included additional keywords such as acquired/secondary resistance, cytotoxicity, leukemia and AML.

Point 3: Introduction needs to be more detailed and give informative background on the subject, particularly the earlier studies that investigated azacitidine resistance.

Response 3: Thank you for your suggestion. We have now modified introduction to provide more details of earlier studies (page number 2, line numbers 61 to 70) and it read asDue to the hydrophilic nature of AZA, transport across the plasma membrane by simple diffusion is limited. Solute carrier (SLC) transporters including SLC28 and SLC29 are well known cellular transporters of multiple nucleoside and nucleoside analogues [10], and therefore were considered potential candidates for AZA transport. However, role of these transporters in AZA resistance is actively debated, as some studies support the role of SLC29A [11-13] while others refute it [14-19]. Similarly, the role intracellular metabolic pathway such as uridine-cytidine kinase (UCK) [13-15, 17, 20-22] and cytidine deaminase (CDA) [15, 16, 18]  remains controversial. This study aims to provide an insight into the mechanism of acquired AZA resistance”.

Point 4: Results: all introductory paragraphs should be transferred to the discussion section (E.g. lines 63-66, 86-88..etc).

Response 4: Thank you for your suggestion. We have modified most of them.  As per your comments we have now deleted/edited line numbers:

  • 108-111
  • 97-98
  • 152-153
  • 168-169
  • 182-183
  • 191

However, we believe that introductory sentences are helpful to comprehend some result sections such as line numbers:

  • 73-74
  • 108-110
  • 124-129
  • 199-200
  • 221
  • 229-231
  • 245-247

Point 5: Line 385: clarify the tested concentrations.

Response 5: Thank you for your valid point. We have now included the tested concentrations in the manuscript (page 11, line 388-389).

Point 6: There is a problem with using abbreviations throughout the manuscript. The full term should be mentioned first with the abbreviation between paresis then the abbreviations should be exclusively used throughout the manuscript. E.g., Line 20: AML should be presented as acute myeloid leukemia (AML), then the abbreviation should be used further. Such errors have been repeated for many abbreviations throughout the manuscript.

Response 6: We acknowledge your concern. We have now modified the manuscript to use the abbreviations throughout the manuscript with the full term only mention at first time.

Point 7: The writing style should be formal from the third-person perspective. Do not use we or our (E.g. line 25, our study should be the current study).

Response 7: Thank you for valid point. We have edited the manuscript to provide more formal writing style.

Point 8: It is not preferable to begin sentences with abbreviations like AZA in line 67.

Response 8: Thank you for your concern. We have modified the sentence (page 2, line 77).

Round 2

Reviewer 2 Report

-